# Meta Continual Learning on Graphs with Experience Replay

**Altay Unal**                                         *unal21@itu.edu.tr*
*Department of Computer Engineering*
*Istanbul Technical University*

**Abdullah Akgül**                                     *akgul@imada.sdu.dk*
*Department of Mathematics and Computer Science*
*University of Southern Denmark*

**Melih Kandemir**                                     *kandemir@imada.sdu.dk*
*Department of Mathematics and Computer Science*
*University of Southern Denmark*

**Gozde Unal**                                         *gozde.unal@itu.edu.tr*
*Department of Computer Engineering*
*Istanbul Technical University*

**Reviewed on OpenReview:** *https://openreview.net/forum?id=8tnrh56P5W*

## Abstract

Continual learning is a machine learning approach where the challenge is that a constructed learning model executes incoming tasks while maintaining its performance over the earlier tasks. In order to address this issue, we devise a technique that combines two uniquely important concepts in machine learning, namely "replay buffer" and "meta learning", aiming to exploit the best of two worlds. In this method, the model weights are initially computed by using the current task dataset. Next, the dataset of the current task is merged with the stored samples from the earlier tasks and the model weights are updated using the combined dataset. This aids in preventing the model weights converging to the optimal parameters of the current task and enables the preservation of information from earlier tasks. We choose to adapt our technique to graph data structure and the task of node classification on graphs. We introduce MetaCLGraph, which outperforms the baseline methods over various graph datasets including Citeseer, Corafull, Arxiv, and Reddit. This method illustrates the potential of combining replay buffer and meta learning in the field of continual learning on graphs.

## 1 Introduction

Deep learning models have proven to perform successfully at numerous machine learning tasks including classification and regression. Despite their celebrated performances, deep learning models tend to provide poor results when they are expected to learn from a sequence of data or tasks (Li & Hoiem, 2017). This process is called continual learning, which aims to train a deep learning model such that it manages learning different tasks in order while avoiding forgetting the tasks that it has learned previously. In continual learning, a model is trained in a way such that it can be retrained for future tasks while preserving information from earlier tasks. The main challenge of continual learning is catastrophic forgetting (McCloskey & Cohen, 1989; Goodfellow et al., 2013), which causes the model to lose obtained information from earlier tasks, leading to a performance drop in the earlier tasks.

The studies addressing the catastrophic forgetting problem in continual learning can be divided into three groups. The first group comprises the memory based studies which either use examples (Rebuffi et al., 2017; Lopez-Paz & Ranzato, 2017) or generate pseudo samples using the stored information (Lavda et al., 2018;

Atkinson et al., 2018). The second group focuses on parameter isolation from earlier tasks such that the model can preserve its performance on earlier tasks (Mallya et al., 2018; Serra et al., 2018), while the third group is the regularization-based methods which propose an extra regularization term to preserve information from earlier tasks while learning on a new task (Kirkpatrick et al., 2017; Li & Hoiem, 2017). While the main problem of continual learning is catastrophic forgetting, continual learning is also challenging due to the differences in the problem setup.

Generally, continual learning has two different setups: class incremental and task incremental (De Lange et al., 2021). In class incremental learning, the individual classes are presented sequentially (Masana et al., 2022) while only the tasks are presented in task incremental learning (De Lange et al., 2021). Class incremental setup requires a deep learning model to classify across the observed and current classes whereas the task incremental setup requires an indicator to separate tasks since the model predicts classes within each task in the task incremental setup. Class incremental setup is more challenging than task incremental setup since the model has no indicator of which task is tested (Zhang et al., 2022). In addition, class incremental learning becomes more challenging as the number of classes increases and the data concerning the earlier tasks is not provided.

Continual learning aims to adjust to newly incoming tasks without forgetting the older ones. Meta learning has the potential to provide benefit to the continual learning problem as the aim of the meta learning is to provide a model with the capability to generalize itself to new tasks. Meta learning is used for scarce data regimes (Finn et al., 2017) and typically refers to processes in which a model learns how to learn. For instance, meta learning becomes an important method for few-shot learning in which deep learning models have very few samples to train with (Nguyen et al., 2019; Chen et al., 2023). The meta learning paradigm guides the neural network model weights so that the model weights do not fit the optimal parameters solely for a given task. As parameters that are optimal for each task cannot be attained, the model tends to find the set of parameters that can collectively adapt to all observed tasks. The model parameters are calculated for the current task without any update. After that, the loss which is called the meta loss, is calculated with the earlier calculated weights, and the model is updated with the meta loss. This mitigates catastrophic forgetting as well as quick adaptation to new tasks. The meta learning method can also be applied to graph structured data (Zhou et al., 2019; Tan et al., 2022).

Graph structured data is a type of data that consists of vertices and edges. In addition, graph structured data is non-Euclidean as it does not have a specific hierarchy and order (Asif et al., 2021). Hence, graph structured data does not have any certain geometry. In certain real world applications such as citation networks or online social networks, data that are represented by graphs dynamically change and, hence tend to expand continuously. Due to the expansion of the data, two problems may arise: (i) new classes may emerge, and the deep learning model cannot handle the newly arrived classes since the model is not trained for those classes; (ii) on the contrary, if the model is trained with the new classes, they may lose the earlier information. Therefore, continual learning on graph structured data becomes necessary.

Continual learning on graphs has been studied recently (Liu et al., 2021; Zhou & Cao, 2021; Zhang et al., 2022) and it has become an emerging field. While the studies addressing this field has similar approaches such as memory-based (Zhou & Cao, 2021) or regularization-based (Liu et al., 2021), they also consider the properties of the graph structured data as their methods can be affected by different factors of the graph structure (Xu et al., 2020) such as topology and irregularity of the graph. In addition, newly arriving tasks alter the dynamics of the graph, interfering with the learning of the model. Therefore, continual learning on graphs diverges and it is handled concerning both the continual learning paradigm and the characteristics of the graph structured data.

In our research, we fused the best of two worlds on continual learning for the graph data type, merging meta learning and replay buffer in a single method with learnable learning rates for the first time, to the best of our knowledge. The comparison of our method to the algorithmic families that build on replay buffer, and meta learning is given in Figure 1. Our developed method fuses the experience replay & selection method in the ER-GNN (Zhou & Cao, 2021) with the meta learning & learnable learning rates from the LA-MAML (Gupta et al., 2020) to advance on graph continual learning. In the graph continual learning problem, we focus on the task of node classification for the class incremental setup, which requires the model to "remember" earlier

| **Algorithm 1** ER-GNN | **Algorithm 2** LA-MAML | **Algorithm 3** MetaCLGraph |
|---|---|---|
| **for** $t = 1$ $to$ $M$ **do** | **for** $t = 1$ $to$ $M$ **do** | **for** $t = 1$ $to$ $M$ **do** |
|    Calculate loss with $T_t$ |    Calculate weights for $T_t$ |    $\mathcal{G}_{aux} \leftarrow \mathcal{B} \cup T_t$ |
|    **for** $t' = 1$ $to$ $t - 1$ **do** |    Meta loss with calculated weights |    Calculate weights with $T_t$ |
|       Get $T_{t'}$ from $\mathcal{B}$ |    Update learning rates |    Meta loss with calculated weights on $\mathcal{G}_{aux}$ |
|       Calculate loss with $T_{t'}$ | **end for** |    Update learning rates |
|    **end for** | |    Extend $\mathcal{B}$ with $T_t$ samples |
|    Sum Losses from $T_t$ & $\mathcal{B}$ | | **end for** |
|    Extend $\mathcal{B}$ with $T_t$ samples | | |
| **end for** | | |

Figure 1: The algorithms for ER-GNN, LA-MAML, and MetaCLGraph (our approach). $\mathcal{B}$ represents the initially empty replay buffer while $M$ is the number of tasks and $T_t$ is the current task. ER-GNN uses the replay buffer to store examples from tasks while LA-MAML uses meta learning for continual learning. MetaCLGraph fuses both by using meta learning in the training phase while storing examples on the replay buffer.

classes so that the model can avoid catastrophic forgetting. Our approach has outperformed the benchmark methods on various graph datasets.

## 2 Background

Graph Neural Networks (GNN) are constructed to process the graph structured data and their inference problems (Kipf & Welling, 2016). Given a graph $\mathcal{G} = (\mathcal{V}, \mathcal{E})$, where $\mathcal{V} = \{v_i\}_{i=1}^{|\mathcal{V}|}$ denotes the set of nodes and $\mathcal{E} = \{e_{ij}\}_{i,j=1}^{|\mathcal{E}|}$ denotes the edges of the graph, GNNs operate in two phases to compute the output embeddings $\mathbf{h}_v^{(l)}$. First, the embeddings are obtained by multiplying the weight matrices of each layer $W^{(l)}$ with the embeddings calculated from earlier layers $\mathbf{h}_v^{(l-1)}$. Next, the embeddings of the neighboring nodes to node $v$ are aggregated by using an aggregation function, and the resulting neighboring embedding is further aggregated with the embedding of the node $v$ from the earlier layer. Separating the embedding of node $v$ from the neighboring embeddings allows GNNs to preserve the information from node $v$.

GNNs contribute to understand the interactions within adjacency matrix and reflect these interactions onto the features so that tasks related to graph structured data can be completed. GNNs are used for node and graph classification while they are also useful for the link prediction between nodes (Asif et al., 2021; Wu et al., 2022). As the graph structured data in real world often evolve, GNNs require continual learning so that the incoming new knowledge can be learned while preserving the information from earlier tasks.

One of the common solutions to the continual learning problem is to use a replay buffer that stores examples from earlier tasks (Zhou & Cao, 2021), ensuring the model's adaptability for incoming tasks without losing earlier information. Using the replay buffer, the model "remembers" the past tasks.

Meta learning is a method that aims to train a model on a variety of tasks (Finn et al., 2017). Given the task set $T = \{T_1, T_2, \ldots, T_i, \ldots, T_M\}$ where $M$ denotes the number of tasks, the model initially calculates the task $i$'s set of parameters which is denoted as $\phi_i$. This process is called the inner update that is used to find the set of parameters for task $i$. After calculating $\phi_i$, the model weights $\theta$ are updated by calculating the derivative of $\phi_i$ with respect to $\theta$. This allows altering the model weights in the direction of the gradient of the loss calculated accordingly to the current task. The objective function for $\theta$ is given below where $\mathcal{D}_i^{\mathrm{tr}}$ and $\mathcal{D}_i^{\mathrm{ts}}$ represent the training and test set for task $i$, respectively.

$$\min_{\theta} \sum_{\text{task } i} \mathcal{L}\left(\theta - \alpha \nabla_{\phi} \mathcal{L}\left(\theta, \mathcal{D}_i^{\mathrm{tr}}\right), \mathcal{D}_i^{\mathrm{ts}}\right). \tag{1}$$

Meta learning ensures that the model does not converge to the optimal parameters of a single task and preserves the adaptability of the model to incoming tasks. Unlike catastrophic forgetting, meta learning aims to develop a "collective intelligence" that allows the model to perform over all observed tasks.

## 3 Related Work

**Continual Learning** mainly deals with the catastrophic forgetting problem. In order to overcome the latter, three approaches are generally used aiming to maintain the adaptability of the model to incoming tasks without any performance loss. The first approach is to isolate the parameters so that the previously obtained information would not be lost. Important parameters for the current task are determined, and those parameters are isolated such that they would not change with the future tasks, and the model can still perform for that specific task (Serra et al., 2018; Mallya et al., 2018). The second approach is to use a regularization term such that the weight importance can be considered, and the changes in the weights that are important for the past tasks are penalized (Kirkpatrick et al., 2017; Aljundi et al., 2018). The regularization prevents the model weights from fitting the optimal parameters of the current task, and hence from forgetting the old tasks. The third approach is to use an additional memory that maintains some examples from the previous tasks. The training process for future tasks uses certain examples from earlier tasks, hence the model would not lose previously obtained information. The usage of additional memory allows the model to observe examples from earlier tasks while it concurrently observes the current task. As a result, the weights would not distance themselves from those of the earlier tasks. The additional memory is used to store information about the earlier tasks to generate similar samples, or it can store examples from the tasks, and those examples can be observed during future tasks. The idea of additional memory is shown to be effective against catastrophic forgetting (Lopez-Paz & Ranzato, 2017; Zhou & Cao, 2021). Capacity saturation is also another challenge in continual learning (Sodhani et al., 2020). The architecture of the model defines the capacity as the components of the architecture affect learning dramatically (Mirzadeh et al., 2022; Shahawy et al., 2022). A model with fixed capacity saturates as it is kept training on more tasks since the model loses its ability to adapt to incoming tasks. This is called the stability-plasticity dilemma (Mermillod et al., 2013). The same dilemma exists in graph continual learning where for GNN architectures, the learning may become limited as a large number of new classes arrives. Addressing this dilemma requires a specialized focus, and while numerous studies addressed the effects of architecture on continual learning (Huang et al., 2021; Feillet et al., 2023), a similar study on GNNs has yet to be conducted.

**Graph Neural Networks** recently have been used in continual learning on graphs (Wang et al., 2020). Since GNNs consider both the adjacency matrix and the features as mentioned earlier, they are capable of discovering the relationship between the nodes and the knowledge held by the graph. Graph continual learning is a very recent topic (Zhou & Cao, 2021) where GNNs are employed for continual learning of graph-structured data. General continual learning approaches, namely regularization based and replay based methods, are adapted to graph continual learning. Inspired by the work in the continual learning field, replay based methods are developed for graph continual learning, storing samples from earlier tasks (Zhou & Cao, 2021) or generating samples by using the stored representations from earlier tasks (Wang et al., 2022). On the other hand, regularization based methods use an additional term in the loss function to adjust the updates in the model weights. In graph continual learning, regularization based methods (Liu et al., 2021; Chen et al., 2021; Cai et al., 2022; Sun et al., 2023) use a regularization term to handle the issues caused by the properties of the graph structured data such as topology and irregularity. Since graph structured data obtains objects which are connected to each other, the tasks obtained by dividing the graph provide several connections among different tasks. Hence, learning different tasks can improve the performance of the GNN for earlier tasks, as different connections can be discovered with every incoming task, and this can be used as a remedy against catastrophic forgetting. In real-world scenarios, graphs tend to be dynamic and the boundaries between the tasks do not usually exist (Tang & Matteson, 2020). Due to the nature of the real-world scenarios, dynamic graph learning (Rossi et al., 2020; Kazemi et al., 2020) can be applied to capture the dynamics of the real-world graphs. Whereas dynamic graph learning primarily concentrates on capturing the up-to-date graph representations (Huang et al., 2022) and it does not focus on the forgetting problem because the observed data can still be accessed by the model. However, in graph continual learning, the observed data is not accessible once after the model is trained on that task, requiring the model to develop

mechanisms to deal with forgetting and use the obtained knowledge. Due to the nature of the real-world graphs, the GNN model becomes more powerful with continual learning, as the same GNN model can be used for different tasks. This allows us to avoid training separate models for each task, and obtain a single model that can perform all tasks which is in line with the ultimate goal of lifelong learning.

**Memory Mechanism.** The implementation of memory mechanisms proves to be really effective (Knoblauch et al., 2020) as memory mechanisms allow models to revisit the earlier examples during the current task and maintain the performances from earlier tasks. Various types of memory units are explored for continual learning problem. In one approach, the implemented memory mechanism can store examples from earlier tasks such that when the model is learning future tasks, it is also trained on the examples on the memory (Zhou & Cao, 2021; Lopez-Paz & Ranzato, 2017). This allows the model not to differentiate from the learned parameters of the earlier tasks. Another approach stores some representations from each task such that those representations can be used to generate examples belonging to earlier tasks (Rebuffi et al., 2017). The generated early-task examples are also used during the training of the current task. Hence, the model can preserve its performance while learning the current task with this approach.

**Meta Learning** is used in reinforcement learning and few-show learning (Finn et al., 2017) to perform multi task learning on a large set of tasks. Since the aim of continual learning is to adjust the model to newly coming tasks while maintaining its ability to perform in older ones, meta learning becomes useful in continual learning. Meta learning prevents overfitting for a specific task by using the gradients of optimal parameters for each task (Gupta et al., 2020). This approach is effective in mitigating catastrophic learning as it prevents the model from overfitting to a specific task and the learning process of the model would not be stopped.

Our proposed approach, namely Meta Continual Learning for Graphs with Experience Replay (MetaCL-Graph), fuses meta learning with a memory mechanism to increase the efficiency in learning incoming tasks. Focused on the node classification problem in the class incremental learning setting, MetaCLGraph uses a meta learning mechanism to improve the model's ability to store already-seen information while learning the current task and incorporates a replay buffer as the memory mechanism that allows the model to observe samples from earlier tasks while learning a new one. In addition, learnable parameters are introduced to the model weights as learning rates so that the updates of each parameter can be different.

## 4    Method

### 4.1    Problem Setup

The goal is to predict the classes of the nodes for a collection of tasks $T = \{T_1, T_2, \ldots, T_M\}$, where $M$ is the number of tasks. The continual graph problem here entails the model to learn those series of tasks in $T$. For each task $T_i$, we have a training node set $D_i^{tr}$ and a test node set $D_i^{tst}$. Node classification aims to predict the right class for each node, i.e., to classify each node in the test node set $D_i^{tst}$ into the correct class by learning the tasks using $D_i^{tr}$. In our graph continual learning setup, we aim to classify incoming nodes based on early observed classes which is also known as class incremental learning (Masana et al., 2022).

### 4.2    Our Method: MetaCLGraph

MetaCLGraph is our proposed solution for graph continual learning which fuses meta learning and episodic memory through an experience replay mechanism. The algorithm for our method is given in Algorithm 4. Although Zhou & Cao (2021) and Gupta et al. (2020) both inspire the MetaCLGraph, the transition to the latter is not straightforward, as the adaptation of the LA-MAML to graph continual learning while using an episodic memory includes overcoming the following obstacles such as lack of graph nodes batches and utilizing the replay buffer efficiently. The model weights are initially calculated for the current task and later updated using gradients calculated while using both stored examples and current task data. This update is called the meta update and it allows instead of fitting for just the current task $T_i$, to be able to learn the previous tasks along with current task $T_i$. The experience replay is obtained when the model is trained on

---

**Algorithm 4** MetaCLGraph-Detailed Algorithm

---

The model weights $\theta$, learning rates $\alpha$, Graph of current task $\mathcal{G}_t$, Buffer $\mathcal{B}$, number of tasks $M$, learning rate for $\alpha$: $\eta$

1: **for** $t = 1 \ to \ M$ **do**
2:     Join buffer samples with the current task dataset: $\mathcal{G}' = \mathcal{G}_t \cup \mathcal{B}$
3:     **for** $epoch = 1 \ to \ E$ **do**
4:         Inner update with the $\mathcal{G}_t$: $\theta_{fast} = \theta - \alpha^t \cdot \nabla_\theta \ L_t \left( \theta, \mathcal{G}_t \right)$
5:         Learning rate update: $\alpha^{t+1} = \alpha^t - \eta \cdot \nabla_{\alpha^t} \ L_t \left( \theta, \mathcal{G}' \right)$
6:         Meta update with the $\mathcal{G}'$: $\theta = \theta_{fast} - max(0, \ \alpha^t) \cdot \nabla_{\theta_{fast}} \ L_t \left( \theta, \mathcal{G}' \right)$
7:     **end for**
8:     Save samples to buffer $\mathcal{B}$ with coverage maximization
9: **end for**

---

a task and some nodes are selected by using the coverage maximization function (4.2.1). The replay buffer can also be seen as an episodic memory since the buffer stores the nodes from the current task at the end of the training. The replay buffer is constructed in this way so that the finished tasks can be revisited by the model using the buffer. The main reason for using the meta learning with a buffer is to prevent the model weights from positioning close to the optimal parameters for a certain task. Following a continual learning paradigm, MetaCLGraph adjusts the model weights to the incoming tasks. Revisiting the stored sample nodes allows the model to remember the earlier tasks. With this added reminder process, the model avoids adapting to only one task. It is able to distance its parameters from the optimal parameters of the current task and also stays tuned to the earlier tasks.

### 4.2.1 Experience Replay

At the end of the training for each task, the additional memory is updated by selecting some examples from that particular task. The additional memory allows our model to "remember" previous tasks as the model weights are trained for the next task. As the model also observes the selected examples, the experience replay allows our model to preserve knowledge from earlier tasks. However, the selection process for the experience replay is important as the representation of a given task relies on the stored nodes.

During the sample selection, we use the coverage maximization method (Zhou & Cao, 2021) to determine the nodes which can represent their tasks accordingly. Coverage maximization method is used as the selection function to maximize the coverage of the embedding space by finding the distances of nodes to the other class. According to the distances, the nodes are ranked and lower ranked nodes are selected. The equation for the coverage maximization is given in the following.

$$\mathcal{N} \left( v_i \right) = \{ v_j \mid \text{dist} \left( v_i - v_j \right) < d, \mathcal{Y} \left( v_i \right) \neq \mathcal{Y} \left( v_j \right) \} \tag{2}$$

where $\mathcal{N}(v_i)$ is the set of nodes coming from different classes with distance $d$ to node $v_i$, $\mathcal{Y}(v_i)$ and $\mathcal{Y}(v_j)$ are the labels of the nodes $v_i$ and $v_j$, respectively. The distance inside the embedding space is minimized so that the classes can be represented inside the buffer. This reinforces the model in remembering the previous classes as the selected examples in this manner cover their embedding spaces.

The experience replay allows us to construct a merged dataset as illustrated in Figure 2. When the training process for a task is finished, several nodes are selected according to the coverage maximization function and stored inside the experience replay. After that, when a new task arrives, the nodes stored inside the experience replay are merged with the data of the current task, and a merged graph is stored. The merged graph allows us to train our model accurately since the model can keep the information learned earlier while learning the current task.

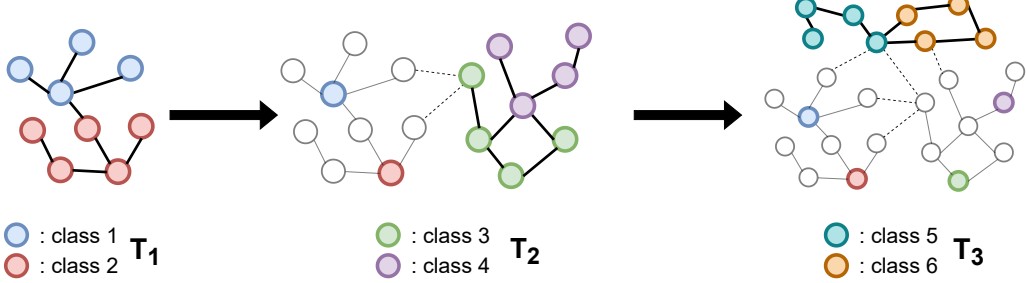

Figure 2: The dataset merging process is illustrated in this figure. The arrows represent the transition between the tasks while coloured nodes represent the active nodes during that task. In our approach, multiple nodes are selected at the end of the task and stored inside the replay buffer. When a new task is arrived, stored nodes are merged with the current task and form a merged dataset. (For interpretation of the references to color in this figure legend, the reader is referred to the web version of this article.)

### 4.2.2 Meta Learning

The meta learning is adapted to MetaCLGraph in two steps as follows. When learning a task $T_i$, the model weights for the current task $T_i$, which are called fast weights $\theta_{fast}$ are calculated using the current task data, shown in Algorithm 4, line 4. This step is called the inner update and allows the model to benefit from the direction of the gradients for the current task. After the inner update, $\theta_{fast}$ is introduced to the model, and the meta loss is calculated when the model is fed with the merged dataset which contains the current task data and the buffer samples from previous tasks. This is called the meta update and the model weights $\theta$ are updated with the gradients with respect to the meta loss using $\theta_{fast}$ in this step, shown in Algorithm 4, line 6. At the end of the meta learning, the model is updated considering the gradients with respect to the current task data and also the samples from previous tasks.

### 4.2.3 Learning Rate Update

In addition to meta learning and experience replay setups, the MetaCLGraph assigns learning rates to every weight, and the assigned learning rates are also updated during the training. Every weight having a different importance for each task during the training as the model learns a task helps in the preservation of the learned knowledge (Gupta et al., 2020).

Using the meta learning loss calculated for the merged dataset, the learning rates for each parameter are also updated by using the gradients. With the updates on the learning rates, the updates on our model weights become more foreseeable. If the model weights require more updates in order to learn the task better, then the learning rate update would be limited. However, if some of the model weights have reached an optimal region, then their learning rate decreases more dramatically than the learning rate of the other weights since the model would tend to preserve that information. This update difference among the model weights allows our model to adjust the parameters according to the tasks, hence the already-acquired information is not lost. For a further detailed reading on how the computation of the gradients with respect to learning rates are calculated, we encourage the readers to refer LA-MAML (Gupta et al., 2020).

## 5 Experiments

The continual learning experimental setups for the GNNs are investigated, and the performance of our setup is compared with the other continual learning setups. The pipeline provided by (Zhang et al., 2022) is used

Table 1: Details of the datasets used in the experiments.

| Dataset | # nodes | # edges | # features | # classes | # tasks |
|---|---|---|---|---|---|
| CiteSeer-CL | 3327 | 9228 | 3703 | 6 | 3 |
| Corafull-CL | 19793 | 130622 | 8710 | 70 | 35 |
| Arxiv-CL | 169343 | 1166243 | 128 | 40 | 20 |
| Reddit-CL | 227853 | 114615892 | 602 | 40 | 20 |

for the experiments. The datasets, baselines, experimental settings, and results are reported and described next. We provide a reference implementation of the proposed model and the experimental pipeline [1].

## 5.1 Datasets

For evaluating MetaCLGraph, four benchmark datasets were employed: Corafull (Bojchevski & Günnemann, 2017), Arxiv (Hu et al., 2021), Reddit (Hamilton et al., 2017), and Citeseer (Sen et al., 2008). Arxiv, Corafull, and Reddit are used in (Zhang et al., 2022) to construct a benchmark for continual graph learning, and Citeseer is additionally included for its significance in graph structured data. In this problem, we divide the datasets into tasks containing two classes for each task. This resulted in 35 tasks from Corafull, 3 from Citeseer, and 20 each from Arxiv and Reddit. For the Reddit dataset, the $41^{st}$ and the last class is dropped since it only has one example. Table 1 provides the detailed task information, and graph structures for each dataset.

To manage device memory requirements, larger graphs like Reddit are divided into batches, while the rest are processed as single batches.

## 5.2 Baselines

The following baselines are used in order to evaluate the MetaCLGraph.

**Base model** is the graph neural network without using any continual learning components. The data is simply passed to the GNN architecture and the model is trained with the provided data. There are no further improvements on the GNN architecture.

**Elastic weight consolidation (EWC) (Kirkpatrick et al., 2017)** focuses on preserving crucial model weights for each task, allowing updates on only less significant weights during the learning of the new tasks to preserve earlier information.

**Memory aware synapses (MAS) (Aljundi et al., 2018)** is a regularization based method that evaluates the importance of the parameters according to the sensitivity of the predictions on the parameters. When new data arrives, the model weights are updated according to the activations of the network so that the important model weights for the newly acquired data can be updated.

**Topology-aware weight preserving (TWP) (Liu et al., 2021)** is a method introduced for graph continual learning which integrates weight preservation considering the graph topology. Graph structure is taken into consideration in order to prevent catastrophic forgetting. After the loss is calculated, it is regularized by calculating the importance score of the model weights according to the topological structure of the provided graph. It benefits from the properties of the graph.

**ER-GNN (Zhou & Cao, 2021)** uses experience replay for this problem. After learning a task $T_i$, sample nodes are saved to the buffer with a selection function and when $T_{i+1}$ is being learned, separate graphs for each learned task from k={0 to i} are constructed, and the GNN is trained with those graphs. The overall

---

[1]https://github.com/ituvisionlab/MetaCLGraph.

loss is calculated with the loss for the current task regularized by the loss calculated from the constructed separate graphs using the experience replay.

**MetaCLGraph (Ours)** uses an episodic memory buffer, and combines meta learning with experience replay.

**Joint** is the method where all the learned tasks are accessed during training. Therefore, it serves as the upper bound for our setup.

EWC and MAS are the techniques focusing on parameter isolation, while ER-GNN uses the additional memory as a replay buffer to store examples from earlier tasks, however, lacking any meta-learning component. TWP is a regularization method that considers the topology of the graph while updating weights. Our model, the MetaCLGraph is compared with those baseline techniques for performance assessment.

### 5.3 Experiment Settings and Performance Scores

The experiments are conducted with a learning rate of 0.005, and each task is trained for 200 epochs. The batch size is selected as 2000 for the batched datasets. Adam optimizer (Kingma & Ba, 2014) is selected as the optimizer. All methods use the graph convolutional network as the backbone GNN architecture (Kipf & Welling, 2016). The selection algorithm relies on coverage maximization. The hyperparameters concerning the compared methods are obtained from the benchmark paper (Zhang et al., 2022) and its repository, as the results in the benchmark are reproducible. The buffer budget is selected as 10 for the replay based methods, namely the ER-GNN and the MetaCLGraph. Given in the benchmark paper (Zhang et al., 2022), the buffer budget sets the number of samples selected for each class in the task, and the entire class or task can be stored when the buffer budget is set high. Setting the buffer budget high does not serve the purpose of continual learning since it would become likely that an entire class or task can be observed in future tasks. To avoid storing an entire class or task that would defeat the purpose of continual learning, the buffer budget is determined as 10. This is based on the observation that all of the datasets contain more than 10 samples for each class. Hence, the buffer budget becomes strict where storing an entire class or task is avoided.

The two evaluation measures are the Average Performance (AP) and the Average Forgetting (AF) (Lopez-Paz & Ranzato, 2017). AP focuses on the model's accuracy in classification of each task. It is obtained by calculating the mean accuracies of the observed tasks so far. AF considers whether or not the model preserves its performance on the observed tasks. The AF is calculated as follows:

$$a_{j,j} - a_{k,j}, \quad \forall j < k \tag{3}$$

where $a_{j,j}$, and $a_{j,j}$ represent the performance scores obtained from the accuracy matrix. $j$ and $k$ represent the numbers of tasks where $k$ represents the current task and $j$ represents a certain old task when it was trained as the current task. In summary, forgetting is determined by finding the performance difference between the last performance and the performance obtained as the current task for each task. All experiments are repeated 5 times on one Nvidia RTX A4000 GPU.

### 5.4 Results

The experimental results are reported in Table 2. The results indicate that the proposed MetaCLGraph outperforms the compared baselines, and achieves the best performance across all datasets in terms the AP measure, and the best performance in terms of the AF measure excluding only the Reddit dataset. These results provide evidence for the fact that the proposed method allows the model to learn the tasks and stores information about the early-observed tasks. The MetaCLGraph outperforms the regularization based methods such as MAS, TWP, and EWC by employing its replay buffer. As mentioned earlier, regularization based methods alter the loss such that the model can preserve the obtained information. However, our proposed method uses the stored examples during the learning of the future tasks as those examples are observed during their training. Therefore, the utilization of the replay buffer allows the MetaCLGraph to outperform regularization based methods. Although MAS has a better AF score for the Reddit dataset, this score is obtained since the AP score is low. Therefore, it can be deduced that the MAS method cannot learn

Table 2: The AP and AF measure results across four datasets for various baselines and the MetaCLGraph. The results are showing mean ± standard deviation across 5 repetitions. Best performances are highlighted in bold.

| Method | Citeseer | | CoraFull | | Arxiv | | Reddit | |
|---|---|---|---|---|---|---|---|---|
| | AP ↑ | AF ↑ | AP ↑ | AF ↑ | AP ↑ | AF ↑ | AP ↑ | AF ↑ |
| Base Method | 31.49±0.15 | -77.48±0.29 | 2.24±0.20 | -94.82±0.33 | 4.85±0.04 | -87.17±1.97 | 5.49±1.05 | -93.39±1.28 |
| EWC | 31.42±0.15 | -77.46±0.32 | 15.02±2.14 | -81.67±2.32 | 4.86±0.01 | -88.35±0.45 | 8.88±1.64 | -94.56±1.75 |
| TWP | 31.37±0.08 | -78.43±0.26 | 16.21±1.76 | -77.81±1.57 | 4.86±0.02 | -88.89±0.13 | 11.75±1.60 | -91.59±1.81 |
| MAS | 31.58±0.07 | -77.47±0.32 | 6.85±1.89 | -88.52±2.14 | 4.86±0.06 | -86.52±0.62 | 12.58±4.67 | **-26.24±8.53** |
| ER-GNN | 46.81±0.33 | -51.10±0.45 | 2.26±0.32 | -95.12±0.40 | 27.83±0.26 | -54.67±0.17 | 37.64±0.64 | -64.55±0.64 |
| MetaCLGraph (Ours) | **51.48±1.64** | **-46.79±2.78** | **64.73±0.34** | **-16.86±0.38** | **31.76±0.74** | **-51.03±0.82** | **67.66±3.64** | -33.01±3.86 |
| Joint (Upper bound) | 76.40±0.20 | - | 81.56±0.14 | - | 46.35±0.85 | - | 98.33±0.26 | - |

the tasks for the Reddit dataset, and its AF score does not mean a better performance as the MAS method has not learned enough information about the observed tasks for storage. The MetaCLGraph outperforms ER-GNN which also utilizes a replay buffer. Although they use the same selection function for the examples to be stored inside the replay buffer, the MetaCLGraph exploits meta learning during the training process. During training, the optimal model parameters are determined for the current task, however, the loss is not calculated until the model observes the stored examples from the earlier tasks. Therefore, the model does not exhibit over-fitting to the current task. On the contrary, it finds the set of parameters that can perform for the earlier tasks while learning the current task. The calculation of the meta loss allows the model to adjust the parameters considering the earlier tasks, hence leads to MetaCLGraph model outperforming the ER-GNN method.

In Figure 3, the performance matrices of the benchmark methods and MetaCLGraph are visualized. The performance scores are presented for all of the tasks after the observation. It can be observed for each dataset that the changes in the performance scores for MetaCLGraph are less pronounced than those for other methods, thanks to a relatively better preservation of the learned information in the former.

## 5.5 Computational Costs

The computational costs are also investigated and the costs are reported in Table 3. The experiments are repeated 5 times on the Corafull dataset. The results show that the computational costs of our proposed model surpass the baseline models. Although the computational costs of our model are the highest among the baselines, its computational costs are justifiable since its performance is beyond other methods and the computational costs are tested on the dataset with the greatest number of tasks among the tested datasets. Corafull dataset contains the greatest number of tasks among the benchmark datasets, showing that the application of our model to other datasets would not be limited since the computational cost difference among the baselines would not be greater.

Table 3: The computational cost table for various baselines and MetaCLGraph.

| Method | Time per experiment (s) |
|---|---|
| Base Method | 159.77 |
| EWC | 381.46 |
| TWP | 418.71 |
| MAS | 177.50 |
| ER-GNN | 234.87 |
| MetaCLGraph | 616.28 |

## 5.6 Ablation

This section focuses on the impact of meta learning and experience replay on the performance of MetaCL-Graph in class incremental learning. MetaCLGraph variants that either exclude meta learning or experience replay are examined to understand their contributions to the model performance. For testing the effects of

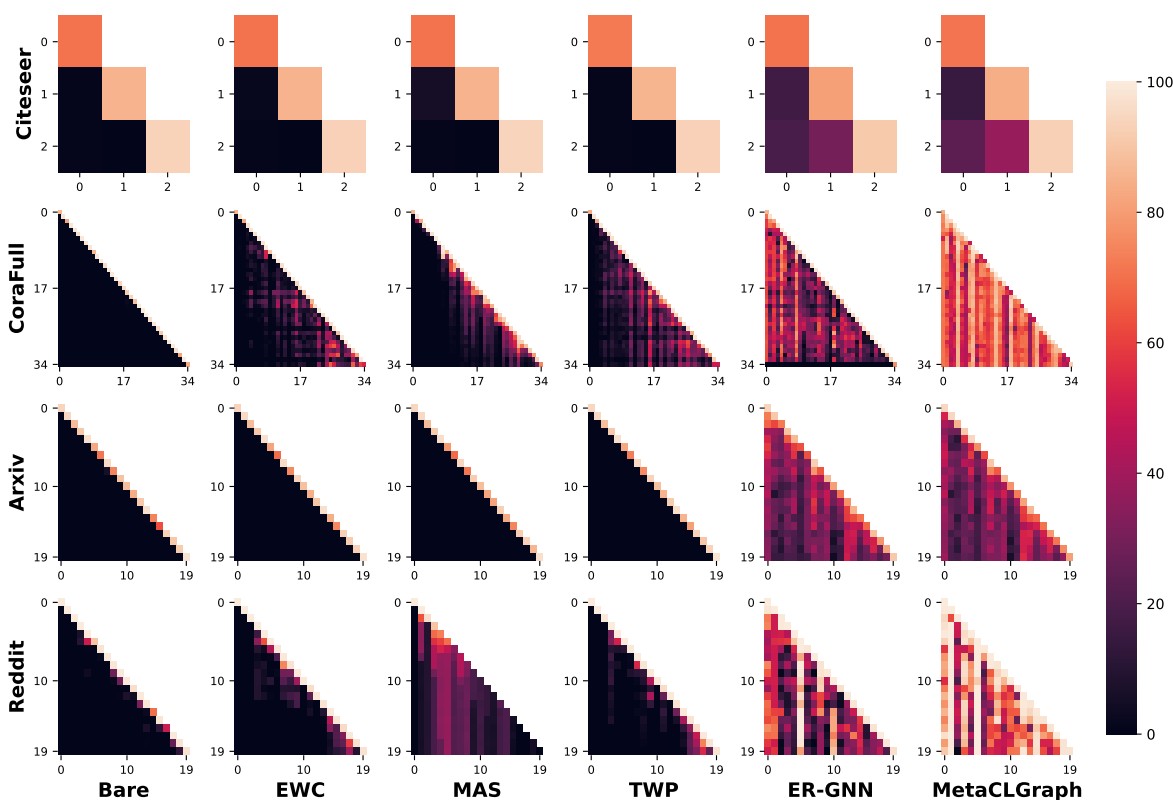

Figure 3: The visualization of performance matrices for MetaCLGraph and benchmark methods for the trained datasets. Each entry in the matrices represents the performance of the method on the column task while learning the task on the row. The determined performance metric on the visualization is accuracy where lighter colours represent the higher score while darker colours represent the lower scores. (For interpretation of the references to color in this figure legend, the reader is referred to the web version of this article.)

experience replay, the buffer budget for the experience replay is changed. When the buffer budget is 0, this would be our approach without experience replay. On the other hand, our method without meta learning would be equivalent to ER-GNN whose results were given in Section 5.4.

The results for the ablation study are given in Figure 4. It can be observed that the model only using meta learning gives similar results to those of the base model. Since this version of our proposed model does not store any information from earlier tasks, and only relies on meta learning, it only learns the current task. Therefore, its AP and AF scores would be similar to the base model since it adjusts its parameters on only the current task and not the other tasks. It is also observed that the number of stored samples increases our model's performance significantly. The number of samples for each class is determined as 10 in order to avoid storing an entire class or task inside the buffer. The results show that when the capacity of the buffer is increased, the performance is improved. However, increasing the capacity too much may cause storing tasks, and storing the tasks would not serve the purpose of continual learning. Although some examples were stored inside the buffer, continual learning setup requires the continuous data flow from earlier tasks to be stopped once the training for a task is finished since the finished tasks become unreachable after the training. However, when the tasks start to be stored in the replay buffer, the model will be retrained on the already visited tasks. Therefore, the continual learning setup is no longer valid, and the model would be trained in a traditional supervised learning setup. This is observed in average forgetting as well. Since all examples from a class may have been stored with a high buffer budget, the model is trained on a task more than once. Therefore, the average forgetting rate becomes greater than zero due to the fact that it has been trained more than specified epochs and the model's performance on that task is improved by meta learning

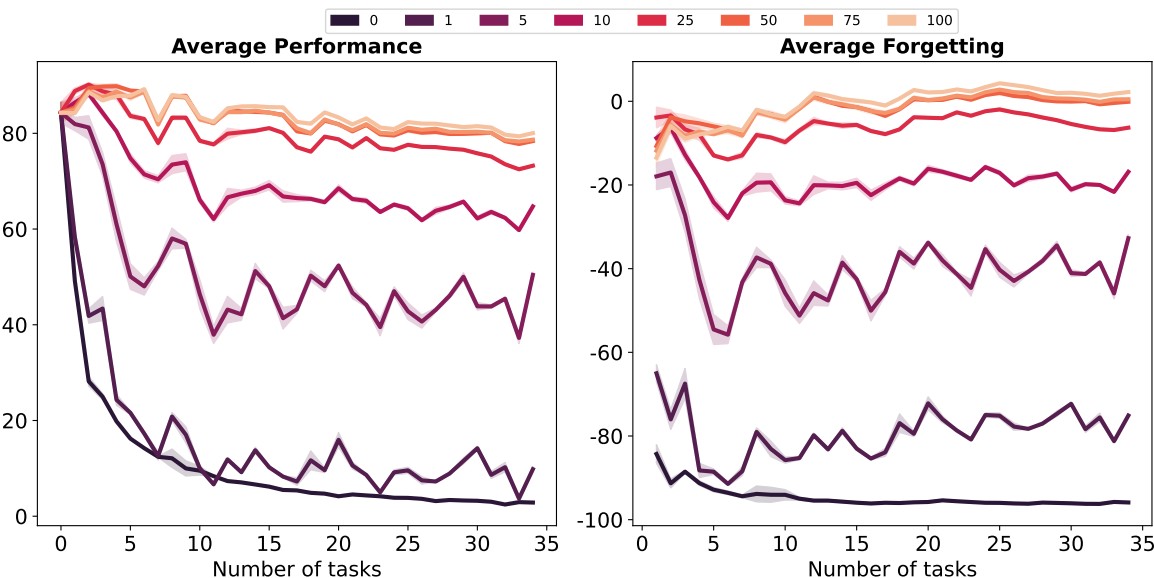

Figure 4: The effect of buffer budget on MetaCLGraph is shown for AP (left) and AF (right). As the buffer budget increases, both the model performance and forgetfulness are improved due to the increase in the stored number of samples per class. (For interpretation of the references to color in this figure legend, the reader is referred to the web version of this article.)

and additional epochs. Therefore, a limit on the number of stored examples in the buffer is imposed so that the integrity of continual learning can be maintained.

## 6   Conclusion

**Summary.** We present the MetaCLGraph for graph continual learning which merges the meta learning paradigm with the use of a replay buffer. MetaCLGraph is compared with the baseline continual learning methods and graph continual learning methods. The experiments provide evidence to our hypothesis that the meta learning paradigm improves the efficiency of the replay buffer and mitigates the catastrophic forgetting problem by conserving the obtained information from earlier tasks. It is observed that the MetaCLGraph outperforms the corresponding baselines in terms of average performance and average forgetting measures.

**Broad Impact.** MetaCLGraph framework can be used in applications for expanding networks where new members and connections are emerging and their characteristics are discovered. The replay buffer of our model can be altered with a different type of memory that store the class representations instead of class examples in order to address data security concerns.

**Future Work.** Our work has shown that the utilization of meta learning improves the efficiency of using a memory mechanism such as a replay buffer. Expanding from MetaCLGraph, a future research direction could be changing the memory mechanism from a replay buffer to a subconscious memory that stores representations for every observed class.

**Limitations.** Storing examples from earlier tasks creates a dilemma for the memory based solutions in terms of data sharing. Our method performs storing a limited number of examples, hence, limiting the buffer capacity and not storing the entire incoming task or class. By limiting the number of examples stored in the replay buffer, our method serves well for the purposes of continual learning while alleviating data privacy and ethical concerns. Another limitation is the increase in the number of tasks as it may limit the application of the proposed model since the overall complexity of the model is already higher than its counterparts.

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
