# OpenReview forum: "Meta Continual Learning on Graphs with Experience Replay"
_TMLR — Accepted by TMLR_

### Review · Reviewer_gQoq · 2023-08-21

**Summary Of Contributions:**

In this work, the authors presented MetaCLGraph for graph continual learning which combines meta learning and experience replay techniques. By using experience replay / a memory module, the proposed model is able to mitigate the impact of catastrophic forgetting which is a major challenge in continual learning. With meta learning, fast prototype of model weights are first computed on the new task without any updates and then using a meta loss to learn a joint representation for previous tasks. This work targets the node classification task for the class-incremental learning setting on graphs. The contributions can be summarized as follows:

- combining ideas from replay buffer and meta learning to form a new learning method for graph continual learning

- strong empirical performance on current graph continual learning benchmark datasets.

**Audience:**

Yes

**Broader Impact Concerns:**

There is no concerns on the ethical implications of the work. The authors also addressed this in the broader impact section of the paper.

**Claims And Evidence:**

Yes

**Requested Changes:**

Following on the weaknesses, here are my requested changes to the paper, [important] means it is critical to my recommendation for acceptance, [minor] means it will improve the paper:

1. [important] the authors briefly mentions the dynamic nature of real world graphs (or dynamic graphs), in Section 3. "In real-world scenarios, graphs tend to be dynamic and the boundaries between the tasks do not usually exist". How is class-incremental learning on graphs different from classification on temporal graphs? Would it be more natural to learn continuously on temporal graphs as new classes naturally arises over time rather than the current setup of splitting the classes manually into different tasks? I think some discussions on this point would be helpful. I will also provide some references for temporal graphs[1,2,3].

[1] Huang, Xuanwen, et al. "Dgraph: A large-scale financial dataset for graph anomaly detection." Advances in Neural Information Processing Systems 35 (2022): 22765-22777.

[2] Rossi, Emanuele, et al. "Temporal graph networks for deep learning on dynamic graphs 2020." URL https://arxiv. org/abs (2006).

[3] Kazemi, Seyed Mehran, et al. "Representation learning for dynamic graphs: A survey." The Journal of Machine Learning Research 21.1 (2020): 2648-2720.

2. [important] In the related work, the authors only discussed the challenge of catastrophic forgetting in continual learning. However, this is not the only challenge, another significant challenge is capacity saturation where the architecture of a neural network needs to continue to grow / adapt when more information arrives otherwise it would be limited in learning by the fixed architecture. This is discussed in various previous work, including [1,2,3]. The same challenge exists in graph continual learning where if the GNN architecture is fixed, then it is possible that its learning may become limited as large number of new classes arrives.

[1] Huang, Shenyang, Vincent Francois-Lavet, and Guillaume Rabusseau. "Understanding Capacity Saturation in Incremental Learning." Canadian Conference on AI. 2021.

[2] Shahawy, Mohamed, Elhadj Benkhelifa, and David White. "A Review on Plastic Artificial Neural Networks: Exploring the Intersection between Neural Architecture Search and Continual Learning." arXiv preprint arXiv:2206.05625 (2022).

[3] Mirzadeh, Seyed Iman, et al. "Architecture matters in continual learning." arXiv preprint arXiv:2202.00275 (2022).

3. [minor] "it allows our model not to fit the optimal parameters
for a given task". It seems counter-intuitive to not want model to find the optimal parameters for a given task. The authors should explain this part better in Section 4.2. The goal of continual learning is to find an optimal parameter set for the joint set of tasks at time t.

4. [minor] the explanation in Section 4.2.2 about the model training on the merged dataset is very confusing. Is theta_fast trained with only data from the new task or the merged dataset? how are the merged dataset utilized to compute loss?

**Strengths And Weaknesses:**

First, I will list the strengths of this work:

1. The authors presented MetaCLGraph, a combination of replay buffer and meta learning. It is interesting to combine these two previous ideas and shows a new approach for graph continual learning.

2. MetaCLGraph showed strong empirical performance on the node classification task across four graph datasets in the class-incremental learning setting based on the CGLB benchmark.

3. Extensive ablation studies are conducted to validate the components in this approach as well as effect of number of tasks.

Now I will discuss some weaknesses and rooms for improvement (more details see requested changes):

1. the related work discussion on continual learning is incomplete. Outside of catastrophic forgetting, continual learning also faces the challenge of capacity saturation which was pointed out in various previous work. This discussion should be added.

2. missing discussion on how graph continual learning is related to another research area: dynamic graph or temporal graph learning. The idea of adding more nodes / classes over time is exactly the setting seen in dynamic graphs and some discussions should be added.

3. some parts of the writeup seems confusing and counter-intuitive which needs improvement

---

> ### Author Response · Authors · 2023-10-02
> **Response to Reviewer gQoq-Part 1**
>
> We thank the reviewer for their constructive feedback. We addressed the reviewer’s concerns and revised our manuscript. Our answers to these concerns are as given below.
>
> **[important] the authors briefly mentions the dynamic nature of real world graphs (or dynamic graphs), in Section 3. "In real-world scenarios, graphs tend to be dynamic and the boundaries between the tasks do not usually exist". How is class-incremental learning on graphs different from classification on temporal graphs? Would it be more natural to learn continuously on temporal graphs as new classes naturally arises over time rather than the current setup of splitting the classes manually into different tasks? I think some discussions on this point would be helpful. I will also provide some references for temporal graphs[1,2,3].**
>
> We thank the Reviewer for their suggestion. We have refined the Related Work section to specify the distinctions between temporal graph learning and graph continual learning more clearly.
>
> *Due to the nature of the real-world scenarios, dynamic graph learning ((Rossi et al., 2020; Kazemi et al., 2020)) can be applied to capture the dynamics of the real-world graphs. Whereas dynamic graph learning primarily concentrates on capturing the up-to-date graph representations (Huang et al., 2022), it does not focus on the forgetting problem because the observed data can still be accessed by the model. However, in graph continual learning, the observed data is not accessible once after the model is trained on that task, requiring the model to develop mechanisms to deal with forgetting and use the obtained knowledge. Due to the nature of the real-world graphs, the GNN model becomes more powerful with continual learning, as the same GNN model can be used for different tasks. This allows us to avoid training different models for different tasks, which is in line with the ultimate goal of lifelong learning.*
>
> **[important] In the related work, the authors only discussed the challenge of catastrophic forgetting in continual learning. However, this is not the only challenge, another significant challenge is capacity saturation where the architecture of a neural network needs to continue to grow / adapt when more information arrives otherwise it would be limited in learning by the fixed architecture. This is discussed in various previous work, including [1,2,3]. The same challenge exists in graph continual learning where if the GNN architecture is fixed, then it is possible that its learning may become limited as large number of new classes arrives.**
>
> Thank you for drawing our attention to this aspect. We recognize the importance of the capacity saturation problem and have added a more detailed explanation on this problem in the Related Work section. The new discussion for capacity saturation in Section 3 is given as follows:
>
> *Capacity saturation is another challenge in continual learning (Sodhani et al., 2020). The architecture of the model defines the capacity as the components of the architecture affect learning dramatically (Mirzadeh et al., 2022; Shahawy et al., 2022). A model with fixed capacity saturates as it is kept training on more tasks since the model loses its ability to adapt to incoming tasks. This is called the stability-plasticity dilemma (Mermillod et al., 2013). The same dilemma exists in graph continual learning where for GNN architectures, the learning may become limited as a large number of new classes arrives. Addressing this dilemma requires a specialized focus, and while numerous studies addressed the effects of architecture on continual learning (Huang et al., 2021; Feillet et al., 2023), a similar study on GNNs has yet to be conducted.*

---

> > ### Author Response · Authors · 2023-10-02
> > **Response to Reviewer gQoq-Part 2**
> >
> > **[minor] "it allows our model not to fit the optimal parameters for a given task". It seems counter-intuitive to not want model to find the optimal parameters for a given task. The authors should explain this part better in Section 4.2. The goal of continual learning is to find an optimal parameter set for the joint set of tasks at time t.**
> >
> > Thank you for your attention. Our proposed model aims to find the optimal set of parameters for all the tasks observed at time t. Therefore, the model focuses on achieving a balance between all the observed tasks rather than optimizing for the current task. According to Reviewer’s feedback, Section 4.2 is revised as follows:
> >
> > *The model weights are initially calculated for the current task and later updated using gradients calculated while using both stored examples and current task data. This update is called the meta update and it allows instead of fitting for just the current task $T_i$, to be able to learn the previous tasks along with current task $T_i$.*
> >
> > **[minor] the explanation in Section 4.2.2 about the model training on the merged dataset is very confusing. Is theta_fast trained with only data from the new task or the merged dataset? how are the merged dataset utilized to compute loss?**
> >
> > $\theta_{fast}$ is calculated only with the data from the current task. It is used to consider how the model weights would change with the current task. After the calculation, $\theta_{fast}$ is given to the model, and the model is “trained” with the merged dataset that contains the current task data and the stored examples from the previous tasks, and the update loss is computed. The update loss is used to update the original model weights. Thus, the model weights are updated after computing the loss with the merged dataset.

---

> > > ### Comment · Reviewer_gQoq · 2023-10-09
> > > **Response to Author Comments**
> > >
> > > I thank the authors for updating their manuscript and adding more discussion on temporal graphs and capacity saturation. My concerns are addressed in the revision.

---

> > > > ### Author Response · Authors · 2023-10-18
> > > >
> > > > Thank you for your comments and your attention.

---

### Review · Reviewer_s6gH · 2023-09-08

**Summary Of Contributions:**

This paper proposes to model consisting of experience replay and meta learning for the continual learning problem. Also, the learning rate of each parameter is gradually decreased according to the gradient on the parameter.

**Audience:**

Yes

**Claims And Evidence:**

Yes

**Requested Changes:**

All the concerns mentioned above should be addressed to improve the paper.

**Strengths And Weaknesses:**

There are multiple concerns about this paper, including some major ones.


1. The so-called meta learning in this work is not real meta learning. It is simply optimizing the models with two steps: 1. optimize with the current task data. 2. update the model again with the extended dataset consisting of both current task data and stored data. While meta-learning is learning to learn and typically find an optimal starting point for the model so that the model can easily adapt to any task. Therefore, the motivation of this 'meta-learning' operation is questionable, why not directly train the model on the extended dataset containing both the current task data and the stored data, so that the model optimization considers both new and old data?

2. Continual learning on graphs has been extensively studied recently, but the related work does not reflect this and only mention few works on this, only Zhou&Cao is mentioned. I would recommend the authors to carefully investigate the recent related works.

3. The writing can be significantly improved. For example, some sentences are very redundant, like the snetences in 4.2.2: ' First, the model is trained
with Dtri in order to determine the parameters for the current task Ti. The parameters for the current task are determined in order to find the direction of the optimal parameters for that task. In order to achieve that, the model weights are calculated and stored.' All these three sentences are talking about one same thing.

4. The so-called meta learning in this work is not real meta learning. It is simply optimizing the models with two steps: 1. optimize with the current task data. 2. update the model again with the extended dataset consisting of both current task data and stored data. While meta-learning is learning to learn and typically find an optimal starting point for the model so that the model can easily adapt to any task.

5. In line 5 of Algorithm 4, how is the gradients with respect to the learning rate alpha calculated?

6. Nodes are stored in the memory. However, when replaying them, does GNN only take in single nodes? This is weird because GNNs will aggregate information from a neighborhood containing multiple nodes (message passing). If one single node is stored, its neighbors are not stored and the message passing cannot work.

7. If the learning rates monotonically decreases, the proposed model will be less and less capable to adapt to new tasks.

---

> ### Author Response · Authors · 2023-10-02
> **Response to Reviewer s6gH-Part 1**
>
> We thank the reviewer for their constructive feedback. We addressed the reviewer’s concerns and revised our manuscript. Our responses are given below.
>
> **The so-called meta learning in this work is not real meta learning. It is simply optimizing the models with two steps: 1. optimize with the current task data. 2. update the model again with the extended dataset consisting of both current task data and stored data. While meta-learning is learning to learn and typically find an optimal starting point for the model so that the model can easily adapt to any task. Therefore, the motivation of this 'meta-learning' operation is questionable, why not directly train the model on the extended dataset containing both the current task data and the stored data, so that the model optimization considers both new and old data?**
>
> We thank the Reviewer’s remarks and we would like to clarify this issue. Our model is built upon LA-MAML, which is itself an adaptation of meta-learning for continual learning. Furthermore, we have extended the application of LA-MAML to handle graph-structured data effectively. In our proposed model, model weights are only updated exclusively during the meta-update step which considers the current task data and buffer samples. Thus, there is not a two-step optimization procedure. The model benefits from both current task gradients and the extended dataset gradients in MetaCLGraph. If the model were only trained with the extended dataset, after a certain number of tasks, the buffer samples may dominate the current task data, causing the model not to learn the current task properly. To avoid that, the gradients for the current task are utilized and continual learning can be maintained.
>
>
> **Continual learning on graphs has been extensively studied recently, but the related work does not reflect this and only mention few works on this, only Zhou&Cao is mentioned. I would recommend the authors to carefully investigate the recent related works.**
>
> We extended our related work and added several studies on the field such as (Zhou & Cao, 2021; Wang et al., 2022; Cai et al., 2022; Chen et al., 2021) .
>
> **The writing can be significantly improved. For example, some sentences are very redundant, like the sentences in 4.2.2: ' First, the model is trained with Dtri in order to determine the parameters for the current task Ti. The parameters for the current task are determined in order to find the direction of the optimal parameters for that task. In order to achieve that, the model weights are calculated and stored.' All these three sentences are talking about one same thing.**
>
> Thanks for pointing this out. We corrected those redundancies. We went through the whole text of the manuscript and conducted a major re-write to significantly improve the overall writing.

---

> > ### Author Response · Authors · 2023-10-02
> > **Response to Reviewer s6gH-Part 2**
> >
> > **In line 5 of Algorithm 4, how is the gradients with respect to the learning rate alpha calculated?**
> >
> > To calculate gradients with respect to the learning rate $\alpha$, we have followed the method proposed by Gupta et al. (2020). If there would be a specific request, we are prepared to adapt and add the gradients derivation to the Appendix.
> >
> > **Nodes are stored in the memory. However, when replaying them, does GNN only take in single nodes? This is weird because GNNs will aggregate information from a neighborhood containing multiple nodes (message passing). If one single node is stored, its neighbors are not stored and the message passing cannot work.**
> >
> > This is a very good observation. Message passing is very essential in GNNs for leveraging the connections between nodes to propagate knowledge and learn structure. We considered the effect of buffer capacity and our proposed model does not perform well when 1 node per class is stored inside the buffer. The coverage maximization function is used for selecting nodes to be stored and if the chosen nodes are interconnected, those connections are preserved to ensure message passing. Thus, the model utilizes the buffer that includes information from the neighboring nodes as the reviewer suggests, as well as it preserves the information from earlier tasks. While our current approach uses coverage maximization to represent the embedding space for the observed classes, there may be some room for improvement considering both the embedding space and the neighbors so that the efficiency of the buffer can be improved.
> >
> > **If the learning rates monotonically decreases, the proposed model will be less and less capable to adapt to new tasks.**
> >
> > Actually, this is a common problem in continual learning. This problem is called capacity saturation, it exists because the architecture of any proposed model has a limit on learning new tasks. We extended our related work section on capacity saturation.
> >
> > To deal with this issue, the learning rate updates in our proposed model preserve the elasticity of our model so that more incoming tasks can be learned. The new discussion for capacity saturation in Section 3 is given as follows:
> >
> > *Capacity saturation is another challenge in continual learning (Sodhani et al., 2020). The architecture of the model defines the capacity as the components of the architecture affect learning dramatically (Mirzadeh et al., 2022; Shahawy et al., 2022). A model with fixed capacity saturates as it is kept training on more tasks since the model loses its ability to adapt to incoming tasks. This is called the stability-plasticity dilemma (Mermillod et al., 2013). The same dilemma exists in graph continual learning where for GNN architectures, the learning may become limited as a large number of new classes arrives. Addressing this dilemma requires a specialized focus, and while numerous studies addressed the effects of architecture on continual learning (Huang et al., 2021; Feillet et al., 2023), a similar study on GNNs has yet to be conducted.*

---

> > > ### Comment · Reviewer_s6gH · 2023-10-11
> > > **Most of my concerns are resolved**
> > >
> > > Thanks for the detailed explanations from the authors and the paper revision. Most of my concerns are resolved and I have changed my rating to leaning accept. However, I still recommend to carefully revise the related work part on continual learning on graph data. This is the most related research areas and many new works have been proposed recently. But in the current version, it seems that only some randomly chosen works are introduced without a clear logic connecting them, and it would be hard for the readers to understand the background of the problem.

---

> > > > ### Author Response · Authors · 2023-10-18
> > > >
> > > > Thank you for your comments and your attention. We have noted your feedback and we will fix the addressed issues.

---

### Review · Reviewer_JeUv · 2023-09-10

**Summary Of Contributions:**

This paper tackles the challenge of continual learning where new classes and tasks continually emerge over time, with the idea of using the reply buffer that can save and learn with the important samples and the meta-learning that can learn model parameters over the distribution of tasks. The authors evaluate the proposed MetaCLGraph on the node classification task of graph-structured data, showing that it can not only improve the performance but also mitigate the forgetting issue.

**Audience:**

Yes

**Broader Impact Concerns:**

The authors clearly discuss the broader impact and limitations of the proposed work in Section 6.

**Claims And Evidence:**

Yes

**Requested Changes:**

Please address the aforementioned weaknesses in marginal contribution against existing works, comparisons against baselines, computational costs, and clarity.

**Strengths And Weaknesses:**

### Strengths
* The idea of using the replay buffer often used in the continual learning setup over the meta-learning framework is interesting.
* The proposed MetaCLGraph significantly outperforms relevant baselines.
* This paper is well-written and easy to follow.


### Weaknesses
* The main weakness is that this work has an incremental contribution against existing works, namely ER-GNN and LA-MAML. In particular, this work directly utilizes the algorithms from both prior works (i.e., using the replay buffer while performing the meta-learning) and does not have clear motivations on why using both of them is necessary.
* The comparison against the LA-MAML model is missing; meanwhile, this LA-MAML, which is the basis model of the proposed MetaCLGraph, is the direct and important baseline to compare.
* The computation costs for performing the meta-learning might be very high especially when the number of tasks is large, in contrast to using other continual learning techniques (e.g., reply buffer or regularization), due to multiple optimizations (i.e., inner and meta updates). It may be beneficial to discuss them.
* The coverage maximization method (Section 4.2.1), which is an important notion when selecting and storing samples in the memory, is not clearly described.

---

> ### Author Response · Authors · 2023-10-02
> **Response to Reviewer JeUv-Part 1**
>
> We thank the reviewer for their constructive feedback. We addressed the issues mentioned in the Weaknesses section.  Our answers and a summary of our revisions could be found below.
>
> **The main weakness is that this work has an incremental contribution against existing works, namely ER-GNN and LA-MAML. In particular, this work directly utilizes the algorithms from both prior works (i.e., using the replay buffer while performing the meta-learning) and does not have clear motivations on why using both of them is necessary.
> The comparison against the LA-MAML model is missing; meanwhile, this LA-MAML, which is the basis model of the proposed MetaCLGraph, is the direct and important baseline to compare.**
>
> Our proposed model represents an adaptation of LA-MAML for graph continual learning. However, the transition from LA-MAML to our method is not straightforward due to the following challenges:
> 1-  LA-MAML relies on having batches for each dataset, which is not inapplicable to numerous graph datasets due to the graph sizes preventing feasible division into batches.
> 2- During training, LA-MAML employs a merged buffer that contains samples selected via reservoir sampling. This causes substantial difficulties when the selected nodes are not adjacent to each other as the message passing cannot be conducted and the stored nodes cannot contribute to the learning.
>
> To effectively address these challenges, in our work, we were inspired by a benchmark paper (Zhang et al., 2022) where coverage maximization was originally used for the implementation of the ER-GNN method, and it was found to be the best selection method after influence maximization in the ER-GNN paper. Thus, we successfully adapted LA-MAML to graph continual learning and utilized a buffer whose aim is to represent the observed classes accurately.
> After the reviewer’s pointer, we added the following text to Section 4.2 in the revised papers to clarify the issue:
>
> *We devise a new method for graph continual learning named MetaCLGraph. We introduce a meta learning solution to continual node classification by incorporating an episodic memory for experience replay. The algorithm for our method is given in Algorithm 4. MetaCLGraph leverages the strengths of both experience replay through a memory buffer (Zhou & Cao, 2021) and meta learning (Gupta et al., 2020) to mitigate the catastrophic forgetting problem. (previously written) Although Zhou and Cao (2021) and Gupta et al. (2020) both inspire the MetaCLGraph, the transition to the latter is not straightforward, as the adaptation of the LA-MAML to graph continual learning while using an episodic memory includes overcoming the following obstacles such as lack of graph nodes batches and utilizing the replay buffer efficiently.*
>
> [Contd.]

---

> > ### Author Response · Authors · 2023-10-02
> > **Response to Reviewer JeUv-Part 2**
> >
> > **The computation costs for performing the meta-learning might be very high especially when the number of tasks is large, in contrast to using other continual learning techniques (e.g., reply buffer or regularization), due to multiple optimizations (i.e., inner and meta updates). It may be beneficial to discuss them.**
> >
> > We thank the Reviewer for their suggestion for a detailed discussion on this matter. We acknowledge this concern and have included a comprehensive computational cost study in the Experiments section of our revised manuscript, providing a thorough analysis of this aspect. The relevant study on computational costs, presented in Section 5.5 (Page 10), is as follows:
> >
> > *The computational costs are also investigated and the costs are reported in Table 3. The experiments are repeated 5 times on the Corafull dataset. According to the results, the computational cost for our proposed model is higher than the baseline models. However, the performance of our proposed model justifies the computational costs as its performance is beyond other methods.*
> > |    Method   | Time per experiment (s) |
> > |:-----------:|:-----------------------:|
> > | Base Method |          159.77         |
> > |     EWC     |          381.46         |
> > |     TWP     |          418.71         |
> > |     MAS     |          177.50         |
> > |    ER-GNN   |          234.87         |
> > | MetaCLGraph |          616.28         |
> >
> > **The coverage maximization method (Section 4.2.1), which is an important notion when selecting and storing samples in the memory, is not clearly described.**
> >
> > Thanks for pointing this out. We revised our discussion in Section 4.2.1 to provide a detailed explanation of the coverage maximization function as follows:
> >
> > *Coverage maximization method is used as the selection function to maximize the coverage of the embedding space by finding the distances of nodes to the other classes. According to the distances, the nodes are ranked, and lower ranked nodes are selected. The equation for the coverage maximization is given in the following.*
> >
> > $$N(v_i)= \\{ v_j \mid \operatorname{dist}(v_i-v_j)<d, Y(v_i) \neq Y(v_j) \\} $$
> >
> > *where $N$($v_i$) is the set of nodes coming from different classes with distance $d$ to node vi, $Y$($v_i$) and $Y$($v_j$) are the labels of the nodes $v_i$ and $v_j$, respectively. The distance inside the embedding space is minimized so that the classes can be represented inside the buffer. This reinforces the model in remembering the previous classes as the selected examples in this manner cover their embedding spaces.*

---

> > > ### Comment · Reviewer_JeUv · 2023-10-06
> > >
> > > Thank you for providing the response to my comments. I acknowledge that I have read the authors'  response. Most of my concerns are resolved and I appreciate that the authors conduct the new experiment on computational costs. On the other hand, I still believe that the main contribution of this work is not significant yet rather more like moderate compared to the existing ER-GNN and LA-MAML.

---

> > > > ### Author Response · Authors · 2023-10-09
> > > >
> > > > Dear Reviewer,
> > > >
> > > > We thank you for your time in reviewing our manuscript and acknowledging our efforts to address your comments and concerns, especially in conducting the new experiment on computational costs.
> > > >
> > > > We understand and respect your perspective on the significance of our work relative to ER-GNN and LA-MAML. However, we would like to offer some further points for consideration:
> > > >
> > > > 1. **Originality and Distinct Contributions**: Our work differentiates from ER-GNN and LA-MAML in certain aspects. While ER-GNN does not utilize the meta learning, LA-MAML is not applicable on graph structured data due to the lack of batches and message passing. Our work combines two techniques used in continual learning separately and merges them in graph continual learning. Although there are similarities, our work shows better performance among different datasets than its counterparts.
> > > >
> > > > 2. **Context of Contribution**: While our work might seem moderate in comparison, its performance is beyond its counterparts. In addition, it utilizes the experience replay with meta learning rather than a regularization and this prevents the forgetting in previous tasks.
> > > >
> > > > 3. **Review Process**: We appreciate the structured feedback process. Your concerns in the first round were perceptive, helping us improve the quality of our work. We handled them as effectively as we could. We understand that the significance of work can be subjective. However, if our revisions align with the criteria in the initial review, we hope that our work can be assessed on its merits.
> > > >
> > > > Lastly, if there are any specific aspects or comparisons with ER-GNN and LA-MAML that you believe could further enhance the significance of our work, we are eager to take them into consideration and make the necessary improvements.
> > > >
> > > > Thank you for your understanding and comments.
> > > >
> > > > Regards,

---

### Review · Reviewer_yCNC · 2023-09-20

**Summary Of Contributions:**

The submission proposes to combine "replay buffer" and "meta learning" to address continual learning problems on graph data. In particular, these techniques are applied to the training of graph neural networks in node classification tasks. With "replay buffer", nodes from previous classes are stored in the buffer for later training. With "meta learning", the model is updated by considering both the current task and previous tasks. The proposed model is applied to multiple graph classification tasks and shows superior performance over several competing methods.

**Audience:**

No

**Broader Impact Concerns:**

No concerns.

**Claims And Evidence:**

No

**Requested Changes:**

While it is hard to suggest any changes to increase the novelty of the work, I would like to see a more rigorous study of the proposed method.

1. Can you evaluate the proposed method in general learning tasks. If better performances are observed across multiple learning domains, the work still deserves a publication. If the proposed model only perform better on graph learning tasks, then a thorough explanation is needed.

2. The writing of the paper should be compressed. The text introduction of the proposed algorithm can be clearer. For example, the following text is hard to understand. The following text quoted from the submission contains examples of writing issues. Questions are in brackets.

"First, the model is trained with Dtri in order to determine (*optimize until convergence?*) the parameters for the current task Ti. The parameters for the current task are determined (*some repetition of the previous sentence*) in order to find the direction of the optimal parameters (*how the "optimality" is defined?*) for that task. In order to achieve that (*what?*), the model weights are calculated (*how? Are model weights the same as parameters?*) and stored. After that, the gradients are calculated (*which gradients, and how are they computed?*) and the model weights (*are they from the stored weights? If not, then how are gradients calculated*), which are called as fast weights θfast, are updated, shown in Algorithm 4, line 4."

**Strengths And Weaknesses:**

Strengths:

1. Graph neural networks with the two strategies are shown to have superior performance in controlled experiments.

Weakness:

1. The work is somewhat incremental as it combines two known techniques.

2. The proposed method is general enough for most learning tasks, but I don't know why it is tested on graph learning tasks. If the authors want to claim the value of the proposed new method, then it should be tested over general learning tasks.

3. The learning tasks in the experiment section seem to be all made-up problems. Could you elaborate more on the practical value of solving these problems? I am also wondering whether there are actual continual learning problems in this field.

4. The writing is a little bit redundant. In particular, some content is repeated (e.g. the discussion of meta learning in sections 2 and 3; the discussion of the meta update at the beginning of section 4.2 and in section 4.2.2). The first three sections can greatly be compressed.

---

> ### Author Response · Authors · 2023-10-02
> **Response to Reviewer yCNC**
>
> We thank the reviewer for the positive and constructive feedback about our manuscript. In accordance with the suggestions of the reviewer, we revised the manuscript. The requested changes were addressed below.
>
> **Can you evaluate the proposed method in general learning tasks. If better performances are observed across multiple learning domains, the work still deserves a publication. If the proposed model only perform better on graph learning tasks, then a thorough explanation is needed.**
>
> Thank you for pointing out the confusion. To clarify this issue, the Introduction section is updated and the following paragraph is added to Page 2.
>
> *Continual learning on graphs has been studied recently (Liu et al., 2021; Zhou & Cao, 2021; Zhang et al., 2022) and it has become an important field. While the studies addressing this field has similar approaches such as memory-based (Zhou & Cao, 2021) or regularization-based (Liu et al., 2021), they also consider the properties of the graph structured data as their methods can be affected by different factors of the graph structure (Xu et al., 2020) such as topology and irregularity of the graph. In addition, newly arriving tasks alter the dynamics of the graph, interfering with the learning of the model. Therefore, continual learning on graphs diverges and it is handled concerning both the continual learning paradigm and the characteristics of the graph structured data.*
>
> Our manuscript focuses mainly on continual graph learning where the prior works are ER-GNN and TWP. The mentioned studies, including ER-GNN and TWP, discuss the continual graph learning problem for large networks such as citation and social media networks. Considering the dynamic structure of such networks, our method alongside the mentioned methods, is specified for graph learning.
>
> **The writing of the paper should be compressed. The text introduction of the proposed algorithm can be clearer. For example, the following text is hard to understand. The following text quoted from the submission contains examples of writing issues. Questions are in brackets.**
>
> **"First, the model is trained with Dtri in order to determine (optimize until convergence?) the parameters for the current task Ti. The parameters for the current task are determined (some repetition of the previous sentence) in order to find the direction of the optimal parameters (how the "optimality" is defined?) for that task. In order to achieve that (what?), the model weights are calculated (how? Are model weights the same as parameters?) and stored. After that, the gradients are calculated (which gradients, and how are they computed?) and the model weights (are they from the stored weights? If not, then how are gradients calculated), which are called as fast weights θfast, are updated, shown in Algorithm 4, line 4."**
>
> Thanks for pointing this out. Alongside some redundancies, we also noted some grammatical and sentence formation errors. After the reviewer’s suggestion, the mentioned section (4.2.2)  can be found on Page 6 as follows:
>
> *The meta learning is adapted to MetaCLGraph in two steps as follows. When learning a task Ti, the model weights for the current task Ti, which are called fast weights θ_fast, are calculated using the current task data, shown in Algorithm 4, line 4. This step is called the inner update and allows the model to benefit from the direction of the gradients for the current task. After the inner update, θ_fast is introduced to the model, and the meta loss is calculated when the model is fed with the merged dataset which contains the current task data and the buffer samples from previous tasks. This is called the meta update and the model weights θ are updated with the gradients with respect to the meta loss using θ_fast in this step, shown in Algorithm 4, line 6. At the end of the meta learning, the model is updated considering the gradients with respect to the current task data and also the samples from previous tasks.*
>
> In addition, the mentioned repetitions about meta learning in Section 3 are also covered and the revised version is given below:
>
> *Meta Learning is used in reinforcement learning and few-show learning (Finn et al., 2017) to perform multi task learning on a large set of tasks. Since the aim of continual learning is to adjust the model to newly coming tasks while maintaining its ability to perform in older ones, meta learning becomes useful in continual learning. Meta learning prevents overfitting for a specific task by using the gradients of optimal parameters for each task (Gupta et al., 2020). This approach is effective in mitigating catastrophic learning as it prevents the model from overfitting to a specific task and the learning process of the model would not be stopped.*

---

> > ### Comment · Reviewer_yCNC · 2023-10-17
> > **Thank you for your responses**
> >
> > Thank you for your responses. You do resolve many of my concerns. However, I still find small writing issues. For example, the first sentence in the abstract is "Continual learning is a machine learning problem": I feel that continual learning is a technique, not a problem. Just like "supervised learning" is another technique or setup.
> >
> >
> > One more question: in Algorithm 4, line 5: is it possible to get negative learning rates here? If so, is there a bad consequence?

---

> > > ### Author Response · Authors · 2023-10-17
> > >
> > > Thank you for your response and your attention. We will fix the remaining issues with the writing as we want to solve all the confusions pointed out by you and the other reviewers.
> > >
> > > **One more question: in Algorithm 4, line 5: is it possible to get negative learning rates here? If so, is there a bad consequence?**
> > >
> > > Although it is possible to obtain negative learning rates in Algorithm 4, line 5, there are not any bad consequences as the meta update is conducted with the learning rate $max(0, \alpha^{t})$, shown in Algorithm 4, line 6. When $\alpha^{t}$ becomes negative, the model stops learning the incoming tasks. This is called capacity saturation and it is another issue in continual learning. The learning rates are updated to increase the elasticity of the model and deal with this issue. We have updated Section 3 by adding more discussion about capacity saturation as follows:
> > >
> > > *Capacity saturation is another challenge in continual learning (Sodhani et al., 2020). The architecture of the model defines the capacity as the components of the architecture affect learning dramatically (Mirzadeh et al., 2022; Shahawy et al., 2022). A model with fixed capacity saturates as it is kept training on more tasks since the model loses its ability to adapt to incoming tasks. This is called the stability-plasticity dilemma (Mermillod et al., 2013). The same dilemma exists in graph continual learning where for GNN architectures, the learning may become limited as a large number of new classes arrives. Addressing this dilemma requires a specialized focus, and while numerous studies addressed the effects of architecture on continual learning (Huang et al., 2021; Feillet et al., 2023), a similar study on GNNs has yet to be conducted.*

---

### Author Response · Authors · 2023-10-02
**Revision of the Submission**

We thank all Reviewers and the Action Editor for their constructive comments. We addressed all the feedback and shared them in OpenReview. We also revised our manuscript according to these reviews, and indicated all changes and restructured parts with “blue” color in both the revised manuscript and the rebuttal.

We improved our work per reviewers’ suggestions. We revised our manuscript to contain a more extensive Related Work section and ensured that our manuscript covers the previous studies on capacity saturation and temporal graph classification. We have also conducted a computational cost analysis to discuss the cost effectiveness of our method among compared methods. In this revised version, the Method section and its subsections are modified to make the ambiguous parts pointed by the reviewers more clear. Finally, the redundancies across the paper are solved to increase the readability of the manuscript.

The bold comments indicate the weaknesses pointed out by reviewers while normal comments are our answers and the italic comments are the quotations from the revised manuscript.

---

### Decision · Action_Editor_aVyh · 2023-10-18

**Recommendation:** Accept with minor revision

**Comment:**

The reviewers generally agree that this is a sound submission, with good experimental results, while they have also found the idea of combining replay buffer with meta learning interesting. However, the paper still has some small issues. I am thus recommending accept with a minor revision, and I request that the final version of the manuscript implements any remaining items promised during the discussion period and carefully considers the final reviewers' comments, particularly the items listed below:

- provide the overall computational complexity of the proposed method, compare it to the complexity of the baselines and discuss whether this could limit the applicability of the proposed method to large scale datasets (Reviewer JeUv).

- revise the related work section on continual learning on graph data to better acknowledge work that has appeared in the past few years and also discuss how the present work is different from closely related works (Reviewer s6gH).

- improve the quality of the presentation and clarity throughout the paper (Reviewers yCNC and s6gH).

**Audience:**

The topic and findings are of interest to some individuals in TMLR's audience, mainly individuals working in continual graph learning and in the more general field of graph representation learning.

**Claims And Evidence:**

The paper proposes a new method for graph continual learning which combines meta learning with an experience replay mechanism. Most claims made in the paper are supported by convincing and clear evidence. For example, the empirical results demonstrate that the use of meta learning indeed improves the performance of the replay buffer and mitigates the catastrophic forgetting problem.